# Quantification of the Coordination Degree between Dianchi Lake Protection and Watershed Social-Economic Development: A Scenario-Based Analysis

**Hansheng Kong** **, Yilei Lu, Xin Dong and Siyu Zeng ***

School of Environment, Tsinghua University, Beijing 100084, China; khs19@mails.tsinghua.edu.cn (H.K.);
luyl16@mails.tsinghua.edu.cn (Y.L.); dongxin@tsinghua.edu.cn (X.D.)
\* Correspondence: szeng@tsinghua.edu.cn

**Abstract:** Dianchi Lake is the largest freshwater lake on the Yunnan–Guizhou Plateau near Kunming City, China. As one of the most polluted lakes in China, although billions of U.S. dollars have been spent trying to clean it up, water pollution and eutrophication are still a bottleneck for regional sustainable development. This research established an integrated approach for the evaluation of the coupling coordination degree to support future planning of the Dianchi Lake basin. Ten future scenarios for possible development directions of Dianchi Lake basin were designed to find the best balance between development and protection. Among these scenarios, a high protection–medium development scenario is the most suitable scenario for future development planning. To further improve the coordination degree, economic growth control and non-point source governance were the most effective and feasible approaches. Furthermore, a water quality model was used to verify the coordination degree. It was found that the high protection–medium development scenario can reach the water quality target in 2025. The coordination degree evaluation could be a practical link to help equilibrate the socio-economic development and environmental protection of the Dianchi Lake basin.

**Keywords:** coordination model; Dianchi lake production; environmental fluid dynamics code (EFDC) water quality model; scenario analysis; policy recommendations; sustainable development

## 1. Introduction

Since the mid-20th century, many countries and regions have witnessed rapid socio-economic development, but this remarkable success has also brought about a disaster for the environment. In order to better solve these problems, the concept of sustainable development has received more attention. Coordination is originally a physics concept. In recent years, it has been adopted much more in the environmental science field, representing a balance between environmental benefit and socio-economic benefit, which is at the core of sustainable development. There are many studies about coordination. Tomal et al. analyzed the coordination degree of socio-economic-infrastructural development in Poland [1]; Ariken et al. performed a coordination analysis between urbanization and the eco-environment in Yanqi Basin [2].

The concept of coordination can be calculated by a coordination degree model. There are multiple coordination models created to solve different coordination problems, such as the membership function coordination degree model, coordination development dynamic change model, coupling coordination degree (CCD) model and dynamic CCD model [3–6]. Among all these models, the coupling coordination degree model (CCD) is the most mature and widely used, which can reflect the intersection of coordinated development among subsystems [7–10]. This model cannot directly simulate a future coordination degree because it needs to be evaluated based on completed events. However, combining it with the scenario analysis method, future coordination trends of the system can still be

calculated [11,12]. Moreover, this model can also reflect which factors in the subsystems have the greatest impact on the development of the subsystem [13,14].

In this research, the Dianchi Lake basin, located in Kunming City, China, is taken as a case study. As one of the most seriously polluted lakes in China, water pollution and eutrophication has become a bottleneck for regional sustainable development. In the past, most studies about Dianchi Lake focused on reducing the pollutant concentration or cutting down the pollutant discharge from a technical perspective [15–17]. This study pays more attention on policy recommendations, combining a coordination model with a scenario analysis to evaluate the degree of coordination between development and environmental protection. On this basis, the environmental fluid dynamics code (EFDC) model is adopted to verify whether the simulated future scenarios can meet the water quality target of Dianchi Lake. The objective of this study is to find the most suitable development scenarios and use these as a reference for the preparation of future development plans of the Dianchi Lake basin.

## 2. Materials and Methods

### 2.1. Study Area

This study uses Dianchi Lake (102°30′–103°00′ E, 24°28′–25°28′ N) as a case study. Figure 1 shows its location. Dianchi Lake is the largest fresh water lake in Yunnan Province, China, which covers 330 km$^2$ and has a 5 m average depth. Waihai is the main part of it, accounting for 96.4% of the total water area [18,19]. In terms of meteorology, the average annual temperature in the Dianchi Lake basin is 15.1 degrees Celsius and the average annual rainfall is 1075 mm. [20]

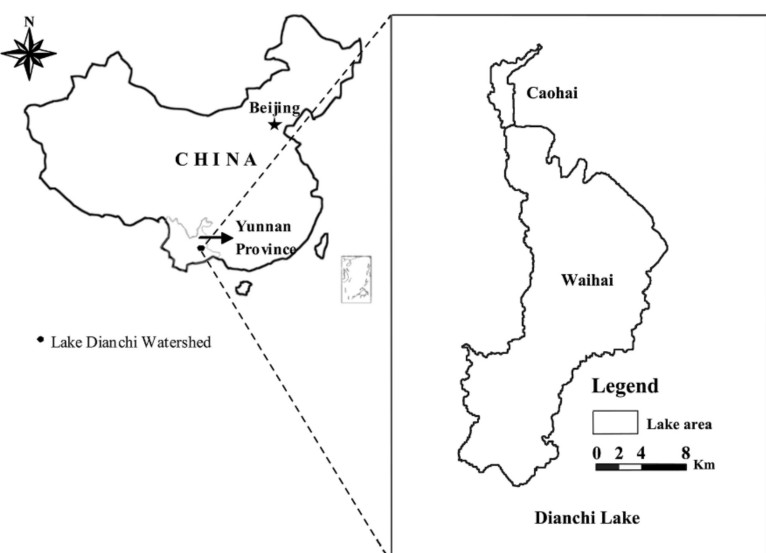

**Figure 1.** Location of Dianchi Lake.

The Dianchi Lake basin is a highly developed region. It only covers 0.75% of the province's land, but nurtures 8% of the province's population and contributes 23% of the province's GDP (gross domestic product) [21]. By the end of 2018, the population of the Dianchi Lake basin had exceeded 4 million with a GDP of 62 billion U.S. dollars, and still has a great chance to grow continuously in the next 10 to 20 years, according to the development plans. It can be said that this area is the most densely populated, economically developed and urbanized area in Yunnan Province. However, at the same time, serious water pollution of the lake is still a huge concern for this area although billions of U.S. dollars have been spent trying to clean it up. According to the latest data in 2018, the water quality rank of Dianchi Lake was still class IV with mild eutrophication, while chemical oxygen demand (COD) and ammonia nitrogen were the main pollutants [22]. Unfortunately, this phenomenon may continue in the foreseeable future.

*2.2. Construction of a Coordination Model*

2.2.1. The Coupling Coordination Degree Model

The coupling coordination degree (CCD) model is the most mature and widely used coordination model, which can reflect the intersection of coordinated development among subsystems. A series of indicators are the basis of it. These indicators are the keys to calculating the development level of subsystems, which are the basic components of the whole system. Table 1 shows all indicators used in this study to evaluate the development level of the socio-economic subsystem and the water environment subsystem. These indicators are all directly derived from the relevant plan of the Dianchi Lake basin, and can best represent the actual development level of subsystems, to ensure the scientificity. Further, the criteria for selecting them are that they should fully reflect the main characteristics of the basin with minimum indicators. Specifically, under the socio-economic subsystem, nine indicators are set to describe three important aspects of the subsystem, which are social development, economic development and industrial development; the water environment subsystem focuses more on indicators related to the current water environment quality in Dianchi Lake. Six indicators are set for it based on three aspects: urban sewage, water environment pressure and water environment carrying capacity.

**Table 1.** Indicators and their weight.

| Subsystem | Aspects | Indicators | Weight | Direction |
|---|---|---|---|---|
| Socio-economic | Social | Population | 0.001 | + |
| | | Population growth rate | 0.004 | + |
| | | Urbanization rate | 0.003 | + |
| | Economic | GDP | 0.17 | + |
| | | Income of urban residents | 0.157 | + |
| | | Income of rural residents | 0.205 | + |
| | Industrial | Agricultural added value | 0.114 | + |
| | | Industrial added value | 0.101 | + |
| | | Services added value | 0.245 | + |
| Water environment | Urban sewage | Drainage per unit of GDP | 0.3 | - |
| | | Sewage treatment rate | 0.027 | + |
| | Water environment pressure | Chemical oxygen on demand (COD) emissions | 0.18 | - |
| | | $NH_3$-N emissions | 0.159 | - |
| | Water environment carrying capacity | COD capacity | 0.167 | + |
| | | $NH_3$-N capacity | 0.167 | + |

In terms of the selection of pollutant indicators, there are three main reasons for choosing ammonia nitrogen and COD as indicators. First, the concentration of phosphorus in Dianchi Lake reached the water quality standards of China in 2011, but the concentration of COD and ammonia still exceed permitted levels. Thus, as a study to provide policy advice, it is necessary to pay more attention to COD and nitrogen, which will still be the key pollutants to be concerned with in the future. Second, the Dianchi Lake basin is surrounded by a chunk of farmland with serious non-point source pollution, and the ammonia nitrogen produced by poultry farming is more or less discharged into the lake without treatment. Therefore, from the perspective of pollutant sources, ammonia nitrogen accounted for the largest proportion of total nitrogen. Third, under the current water environment management system of China, COD and ammonia nitrogen are the most important and widely monitored pollution indicators. Therefore, their emission data are easier to obtain. Conversely, it is much harder to obtain complete phosphorus emission data.

### 2.2.2. Determination of Indicators' Weight

The indicator system should be normalized to eliminate the dimension, by Equation (1):

$$X'_{ij} = \begin{cases} \frac{X_{ij}}{\alpha_j}, & | \ X_{ij} \ is \ a \ positive \ indicator \\ \frac{\beta_j}{X_{ij}}, & | \ X_{ij} \ is \ a \ negative \ indicator \end{cases} \tag{1}$$

where $X_{ij}'$ and $X_{ij}$ represent the standardized and the primitive value of indicator $j$ in year $i$; and $\alpha_{ij}$ and $\beta_{ij}$ represent the standard value of each indicator. Here, different symbols are used to distinguish whether the indicators are positive or negative. If an indicator has a positive impact on the subsystem, such as the growth rate of GDP (gross domestic product), it could be considered a positive indicator. On the contrary, indicators with negative impact such as pollutant emissions are negative indicators.

The weights of indicators represent their importance to the model. Instead of an average weighting method, the entropy weighting method is used to determine the indicators' weight. This method introduces the concept of entropy, regarding indicators with greater variations and fluctuations as having greater impact on the subsystem, while those with less fluctuation have less impact on the subsystem. The weight is set based on the magnitude of it, to avoid the subjectivity of using the experience to determine it. Table 1 also shows the results of the weights, which are calculated by Equations (2)–(5) as follows:

$$P_{ij} = \frac{X_{ij}}{\sum_{i=1}^{n} X_{ij}} \tag{2}$$

$$e_j = -\frac{1}{\text{Ln } m} \sum_{i=1}^{n} P_{ij} \tag{3}$$

$$d_j = 1 - e_j \tag{4}$$

$$\omega_j = \frac{d_j}{\sum_{j=1}^{n} d_j} \tag{5}$$

where, $P_{ij}$ represents the proportion of indicator $j$ in year $i$; $e_j$ represents the entropy of indicator $j$; $d_j$ represents the difference coefficient of indicator $j$; $\omega_j$ represents the weight of indicator $j$; $m$ means the total number of years and $n$ means the total number of indicators.

### 2.2.3. Establishment of the CCD Model

Assume that $x_1, x_2, x_3 \ldots x_p$ represent the indicators of the socio-economic development subsystem, and $y_1, y_2, y_3 \ldots y_q$ represent the indicators of the water environment subsystem. The development index of each subsystem can be calculated by Equations (6) and (7). Among them, $s(x)$ and $e(y)$ represent the development index of the socio-economic development subsystem and the water environment subsystem; $\omega_s$ and $\omega_e$ represent the weight of each indicator in the two subsystems:

$$s(x) = \sum_{s=1}^{p} \omega_s x'_s \tag{6}$$

$$e(y) = \sum_{e=1}^{q} \omega_e y'_e \tag{7}$$

The CCD model can be constructed by Equation (8):

$$C = \left\{ \frac{s(x) \times e(y)}{\left[ \frac{s(x)+e(y)}{2} \right]^2} \right\}^{1/2} \tag{8}$$

The coordination degree *C* indicates the difference between the development levels of the two subsystems ($0 \leq C \leq 1$). Specifically, $C = 1$ means the coupling degree between the two subsystems reaches the peak, while $C = 0$ means completely irrelevant and unco-ordinated. Comparing $s(x)$ with $e(y)$, $s(x) > e(y)$ means the socio-economic development level exceeds the carrying capacity of the water environment, while $s(x) < e(y)$ means the water environment capacity cannot be fully utilized to support the social and economic development.

We defined *T* to represent the overall system development index:

$$T = a \times s(x) + b \times e(y) \tag{9}$$

Assuming that socio-economic development is as important as water environment protection, $a = b = 0.5$.

$$D = \sqrt{C \times T} \tag{10}$$

*D* represents the coordinated development degree, which can simultaneously reflect the coordination and development level. It is possible that some situations with a poor coordination degree will have a higher coordinated development degree. In fact, a little sacrifice on coordination can be accepted if the overall development level is very high [23,24].

### 2.3. Scenario Definitions

The simulation interval of scenario analysis in this study is 2016–2025. City planning is the main reference standard for future development directions and government works. The 13th Five-Year Plan of the Dianchi Lake basin is an important plan composed by the Kunming municipal government in 2015, outlining the expected development direction of the Dianchi Lake basin during 2016 to 2020 [25]. On the basis of this, a reasonable extrapolation is made and a scenario called the "Planning scenario" is set, which indicates that the future development is fully carried out as expected. The values of the other eight scenarios are taken on both sides of the Planning scenario. Each scenario is set by different development speeds of two important aspects: socio-economic development and water environment protection. Specifically, three different socio-economic development speeds are set based on different population, economic growth and industrial structure, and can be referred to as the "High development", "Medium development" and "Low development". Similarly, three types of water environment protection, "High protection", "Medium protection" and "Low protection", are defined based on different levels of water pollutant emissions. The Medium development–Medium protection (M-M) scenario is exactly the Planning scenario mentioned above. In addition, another scenario called the "Historical scenario" is set, which maintains the historical mode and is irrelevant to the city planning.

Each development scenario can represent a possible future development direction of the Dianchi Lake basin. For example, the Low protection–High development (L-H) scenario means the Dianchi Lake basin will incline more resources to socio-economic development in the future, which will bring rapid development, but the cost is a sacrifice to the water environment. The Low protection–Medium development (L-M) scenario indicates that the socio-economic subsystem will grow as expected, but the water environment protection fails to reach the goal. Another example is the High protection–Low development (H-L) scenario, which focuses too much on water environment protection but ignores the development of the economy. Because the purpose of this study is to find the most appropriate scenario according to its coordination degree, most indicators used to set the scenario are the growth rates of those indicators set to build the CCD model (list in Table 1) so that the coordination degree can be calculated easily. In particular, the water use situation is subdivided to distinguish the emissions of different industries. The values of each scenario are shown in Table 2.

**Table 2.** Values of scenario indicators.

| Socio-Economic Development | High | Medium | Low |
|---|---|---|---|
| Population growth rate | 3% | 1.9% | 1% |
| Urbanization growth rate | 0.94% | 0.84% | 0.74% |
| GDP growth rate | 10.50% | 9.50% | 8% |
| Agricultural growth rate | 5.50% | 5% | 4.50% |
| Industrial growth rate | 12.00% | 10.7% | 8% |
| Growth rate of service industry | 10% | 7.5% | 6% |
| Growth rate of urban resident income | 10.50% | 9% | 7.50% |
| Growth rate of rural resident income | 11.50% | 10% | 8.50% |
| **Water Environment Protection** | **High** | **Medium** | **Low** |
| Water use efficiency of primary industry | Increase by 30% | Increase by 12% | Increase by 3% |
| Water use efficiency of secondary industry | Increase by 30% | Increase by 18% | Increase by 5% |
| Water use efficiency of tertiary industry | Increase by 30% | Increase by 12% | Increase by 3% |
| Sewage treatment rate | 98% | 96% | 94% |
| COD emission | Decrease by 18% | Decrease by 10% | Decrease by 2% |
| $NH_3$-N emissions | Decrease by 22% | Decrease by 13% | Decrease by 4% |
| COD capacity | Increase by 10% | Unchanged | Decrease by 10% |
| $NH_3$-N capacity | Increase by 10% | Unchanged | Decrease by 10% |

### 2.4. EFDC Model

In order to further analyze the accessibility of Dianchi Lake protection targets, the EFDC model will be used to simulate the water quality under typical scenarios in 2025. The environmental fluid dynamics code (EFDC) model is a typical three-dimensional water quality model that can simulate the water quality change process by analyzing the different input of pollutants [26]. This model has been widely used in various water bodies, including rivers, lakes, reservoirs and coasts [27–30]. For example, Gong et al. investigated the responses of water quality to different extreme hydrological conditions associated with rainstorms [31]; Arifin et al. used the EFDC model to simulate thermal behavior in Lake Ontario [32]. The Waihai EFDC model used in this paper can simulate the mutual transformation process and reaction of various substances under different situations by changing the parameter values such as the flow of the river into the lake and the concentration of pollutants. This model has been calibrated and verified in past research [33]. Among the input parameters, the flow and temperature of each river are derived from historical data, while the inputs of COD and ammonia nitrogen are obtained from the scenario analysis and need to be divided by time and space. The pollution loads of COD and ammonia nitrogen in the Dianchi Lake basin have two parts: point source and non-point source. The point source mainly comes from the factory and domestic sewage, which is irrelevant to rainfall intensity, so its discharge intensity can be assumed to be evenl throughout the year [34]. The non-point source emission will be affected by different rainfall intensity, so it can be split according to historical monthly rainfall [35]. There are 15 rivers entering the Waihai with great differences in drainage areas and locations, which provides a basis for dividing the pollution load by space [20,36].

## 3. Results and Discussion

### 3.1. Development Index of Each Scenario

Figure 2 shows the trends of subsystem development indexes in different scenarios from 2016 to 2025. Where $s(x)$ represents the development index of the socio-economic subsystem and $e(y)$ means the development index of the water environment subsystem.

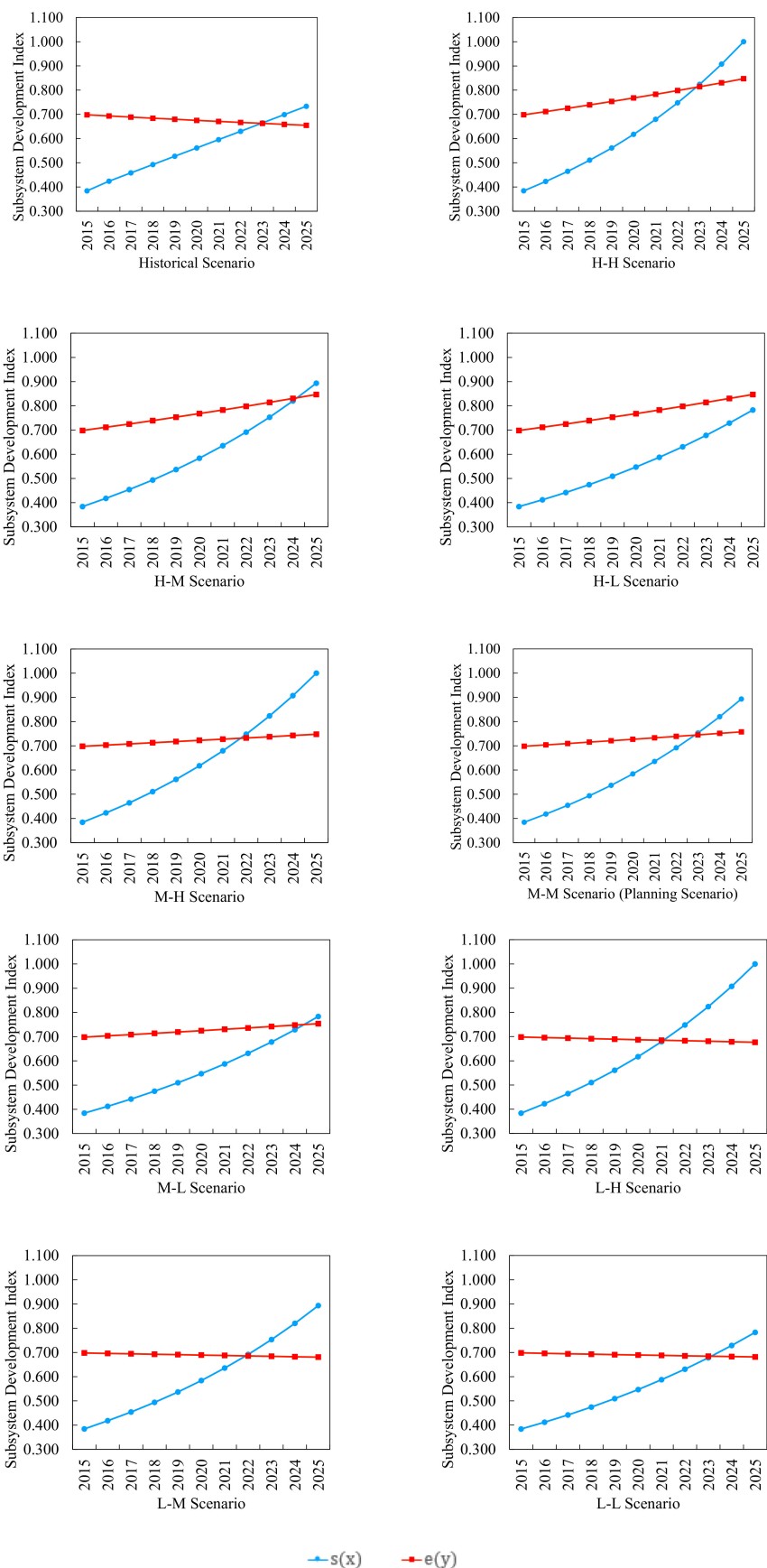

**Figure 2.** Development indexes of each scenario.

The development indexes of the socio-economic subsystem in all scenarios improve significantly, among which the Historical scenario has the smallest increase and the H-H scenario has the largest increase. This trend reflects that the Dianchi Lake basin is still in a relatively rapid development stage. In other words, development is still the top priority in the Dianchi Lake basin, and it is not advisable to sacrifice economic development and completely incline resources to water environment protection. The development indexes of the water environment subsystem ($e(y)$) in almost all the scenarios have a relatively low growth rate during these 10 years. This indicates that insufficient improvement in protection will lead to the deterioration of the water environment system, which cannot be accepted in the future.

### 3.2. Analysis of the Coupling Coordination Degree

The coordination degree $C$ curve is shown in Figure 3, and the coordinated development degree $D$ curve is shown in Figure 4.

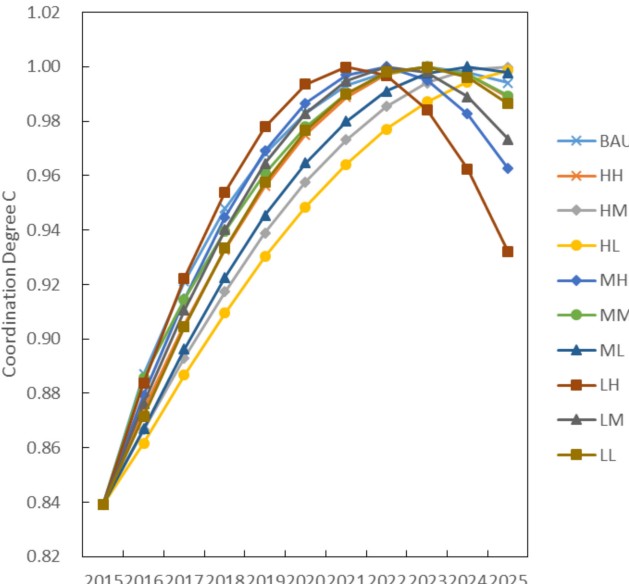

**Figure 3.** Trend of coordination degree C.

The coordination degree of all scenarios increases first due to the increase of $s(x)$ in the near future, and will eventually show a lagging development of the water environment system in the long term because $s(x)$ grows more quickly than $e(y)$ and will eventually exceed $e(y)$. The L-H scenario will reach the turning point first in 2021. If it continues to develop in this way in the future, the coordination degree would fall to 0.7 in five years (general maladjusted system), and to 0.5 in 2032 (serious maladjusted system). Although the L-H scenario will bring certain growth dividends due to its rapid social and economic development in the short term, the huge system imbalance caused by the destruction of the water environment will have an irreparable impact in the long run. Therefore, it is obviously unacceptable to take this scenario as a reference for future planning. Similarly, the L-M and M-H scenarios will reach an inflection point in 2022, and the coordination degree of these scenarios will fall to 0.5 in the next decade after 2025, becoming a serious maladjusted system. Although the gap between the socio-economic subsystem and water environment subsystem in these scenarios is not as large as that in the L-H scenario, the imbalance will appear only three to five years later than the L-H scenario and it will still cause great damage to the system coordination. Therefore, these scenarios are also not suitable for future development decision-making. In addition, it can be found that the COD discharge will exceed the COD capacity of Dianchi Lake after 2024 under the L-H, L-M and L-L scenarios due to the substantial increase of COD discharge and decrease of water environment capacity, which further confirms their disadvantage. Except for these

scenarios mentioned above, none of the other scenarios has seen a significant decline in coordination degree, which means the benefits of their coordinated increase can be fully enjoyed in the foreseeable future. Therefore, from the perspective of coordination degree, these scenarios are all reasonable.

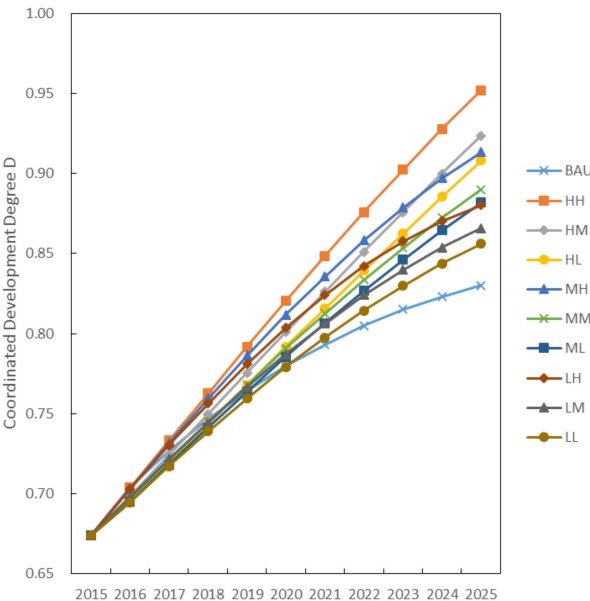

**Figure 4.** Trend of coordinated development degree D.

The coordinated development degrees will all increase during the simulation period, with various speeds. The coordinated development degree can further reflect the overall development speed compared with the coordination degree. Generally speaking, if the coordination degrees of several scenarios were the same, a scenario with a higher coordinated development degree would be a better choice. The coordinated development degree of the Historical scenario grows most quickly at the beginning, but slows down quickly after then. By 2025, it has fallen to the last among all the scenarios, caused by its fast decline in the development index of the water environment subsystem. In other words, under this scenario, the deterioration of the water environment will make the Dianchi Lake basin much more uncoordinated and unbalanced. The L-L scenario is similar to the Historical scenario, so this scenario is also not suitable for future planning.

The H-H, H-M, M-L and Planning scenarios (M-M) all have a good trend of coordination degrees, and the coordinated development degree can basically reach 0.9 in 2025. However, the H-H and M-L scenarios still have some disadvantages. The High protection–High development (H-H) scenario is overfulfilled in both socio-economic development and water environment protection, which is hard to maintain due to the limited human and material resources. The future development plan should follow practical principles, so obviously, this scenario is unable to meet it. The Medium protection–Low development (M-L) scenario means that water environment protection will be carried out successfully, but the socio-economic development speed cannot meet the goals. This scenario may happen, but it is not welcomed for the government because it means a recession caused by some unforeseen reasons.

According to the results of the coordination analysis, the H-M scenario and M-M scenario (Planning scenario) perform best, so these two scenarios are the best reference scenarios for future planning.

*3.3. Identification and Optimization of Key Factors in Coordinated Development*

It can be found from the scenario analysis that in order to improve the coordination degree based on the premise of keeping the coordinated development degree unchanged, the gap between the development index of the social economic subsystem ($s(x)$) and the

water environment subsystem ($e(y)$) needs to be narrowed, which means reducing $s(x)$ or increasing $e(y)$ in the long term. The weight of indicators in the coordination model can reflect their influence on subsystem development index, so those indicators with high weight are the key factors to improve system coordination.

In the socio-economic subsystem, economic growth, residents' income and industrial added value are the key factors to consider. Reducing the growth rate of residents' income is contrary to the government's goal of improving people's livelihood, so this method is unpractical. Reducing the economic growth rate can also control industrial production, which means it is a great method for reducing $s(x)$. Generally speaking, the growth rate of GDP (including the growth rate of added value of each industry) has a marginal effect, that is to say, the growth has a certain limit. At present, Kunming's GDP growth rate is 9%, while the national GDP growth rate is only 6.5% [37]. In the long term, this high-speed development will naturally fade, so it is possible to reduce the economic growth rate by macro-control and the investment of more resources into environmental improvement. Specifically, if the expected GDP growth rate could be reduced to 6.5%, $s(x)$ in 2025 would be reduced by 15%, which would effectively improve the coordination degree.

A more effective way to improve coordination is to increase $e(y)$, which can be effectively achieved in terms of urban sewage, pollutant discharge and water environmental capacity. As for the water environment capacity, several water diversion projects have been implemented in the Dianchi Lake basin, and it is not feasible to further improve water environment capacity through the establishment of new water diversion facilities. Furthermore, upgrading the sewage treatment plant is not cost effective as existing sewage treatment plants in the basin have already met rather stringent discharge standards, which means a rather high cost for further improvement. Therefore, increasing urban non-point source pollution reduction is the most feasible way to increase $e(y)$. In addition, reducing the amount of sewage by improving the wastewater reuse rate is also a good choice.

### 3.4. Water Quality Simulation under Typical Scenarios

This section analyzes the water quality of Dianchi Waihai in 2025 under three typical scenarios: the H-M scenario, M-M scenario (Planning scenario) and Historical scenario, to confirm whether these scenarios can meet the water quality target. There are eight national control points (hereinafter called the "NCPs") in the Dianchi Waihai, and the simulated data of these NCPs are the most direct basis for evaluating the water quality. Figures 5–7 show the COD and ammonia nitrogen concentrations of these NCPs under three typical scenarios.

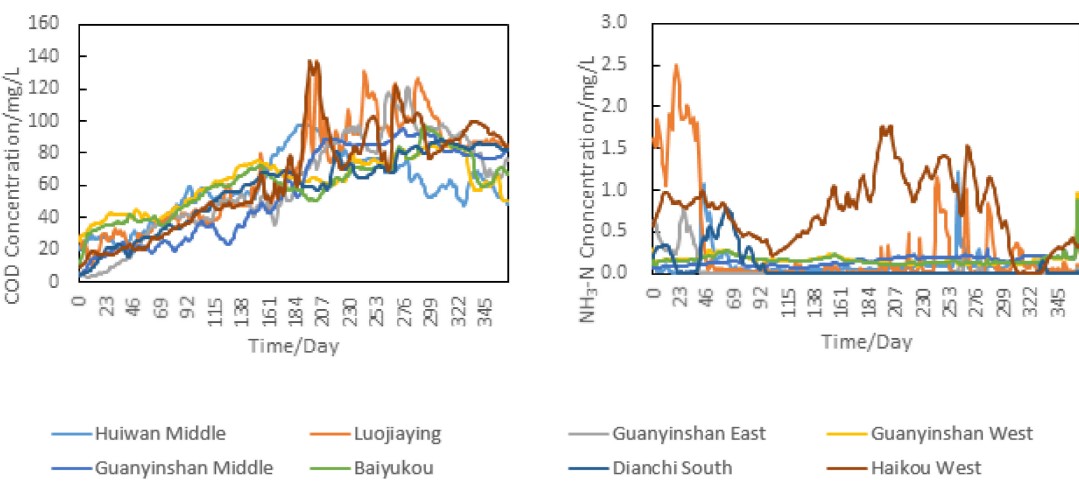

**Figure 5.** COD and ammonia nitrogen concentrations at eight national control points (NCPs) under the Historical scenario in 2025.

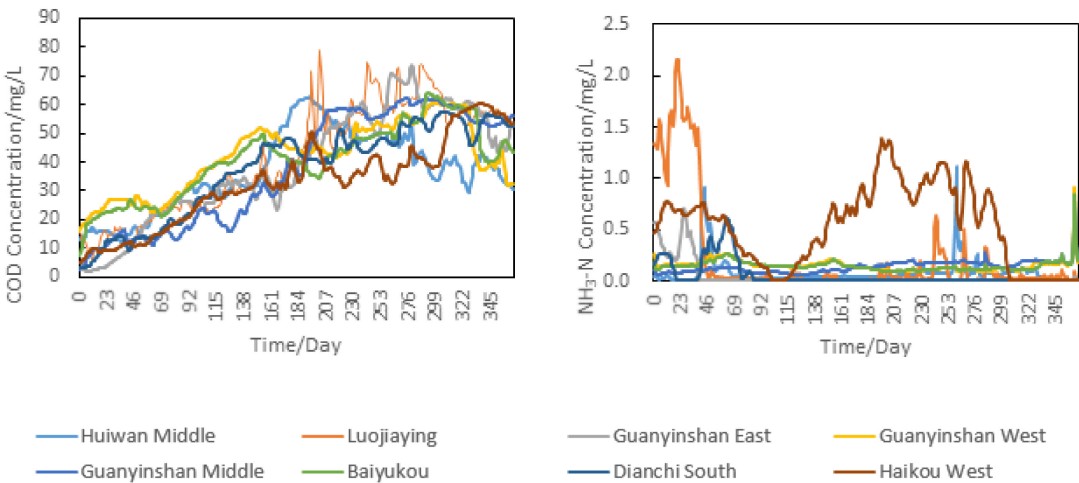

**Figure 6.** COD and ammonia nitrogen concentrations at eight NCPs under the High Protection–Medium Development (H-M) scenario in 2025.

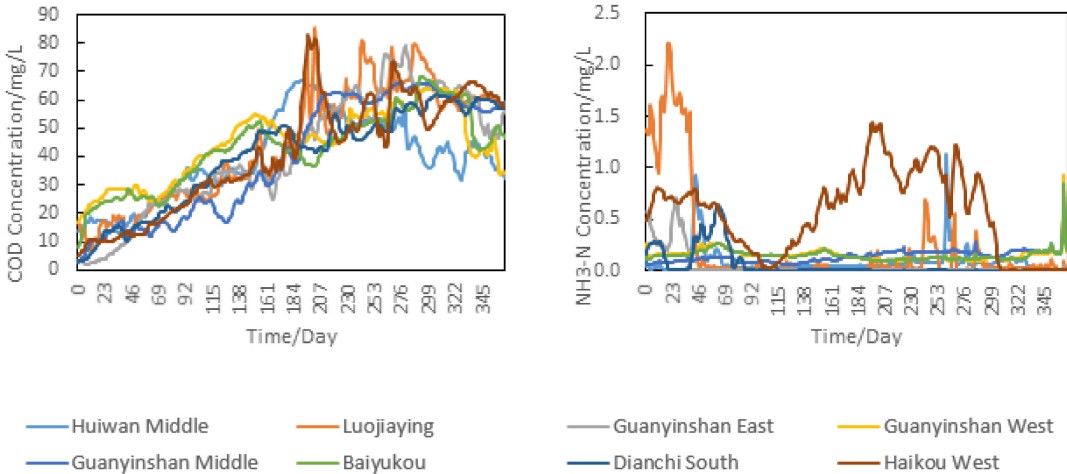

**Figure 7.** COD and ammonia nitrogen concentrations at eight NCPs under the Medium development–Medium protection (M-M) (Planning) scenario in 2025.

Under the Historical scenario, the COD concentration curve can be divided into two stages. The first stage is from the first day to the 200th day. The COD concentration continues to rise and reaches the peak with a maximum value of 130 mg/L, which can be explained by rainfall intensity. Non-point source pollution affected by rainfall intensity accounts for a large proportion of COD emission; therefore, the change trend of COD concentration is similar to that of rainfall intensity, which is lowest in January and highest in July. After the 200th day, the rainfall intensity decreases, and the amount of COD entering the lake falls, which leads to a balance of biochemical reactions and COD concentration. As for spatial distribution, four NCPs located in the middle of the lake have the lowest concentration (about 65 mg/L), while the concentration of the other four points is about 90 mg/L. It can be found that a point far away from the emission source will show a lower concentration due to the dilution and degradation of pollutants during the migration process. According to the monitoring data in 2015, the annual average COD concentration in the Waihai was 48 mg/L, and the maximum value was 62 mg/L [37]. The simulated data of the Historical scenario in 2025 is even worse compared with that.

Under the Historical scenario, the concentration of ammonia nitrogen in the middle of the lake is stable throughout the year, which may be because the ammonia nitrogen will convert rapidly into other forms of nitrogen with the increase of migration distance.

In comparison, the ammonia nitrogen concentration of the NCPs near the main urban area are relatively high, which may be explained by their shorter migration distance. In addition, the highest value of ammonia nitrogen concentration will break the Class V water standard of China, which will also be a huge step back compared with the average value in 2015 (0.2 mg/L). This consequence further confirms that the Historical scenario is not conducive to the development of the Dianchi Lake basin.

The concentration curves of the H-M and M-M scenarios are similar to that of the Historical scenario, with various values. Under the H-M scenario, the peak value of COD concentration will appear in the Luojiaying point (78 mg/L), and it will also briefly exceed 70 mg/L in the Guanyinshan East point. At the end of the simulation period, the COD concentrations in the north and south points are stable at around 50 mg/L and that in the middle points are stable at about 40 mg/L, which will be a great improvement compared with the monitoring data in 2015. As for the ammonia nitrogen concentration, except for the Luojiaying and Haikou West points, all points can achieve the Class III water standard of China all year round, and some can even reach the Class II water standard. The simulation result of the M-M scenario is slightly inferior to that of the H-M scenario, but still much better than the Historical scenario.

Through the analysis of pollutant concentration in the national control points, it can be found that both the H-M and M-M scenarios can improve the water quality of Dianchi Lake, and the H-M scenario performs better. Comparing this conclusion with the previous coordination analysis results, it can be found that the H-M scenario is the most suitable pattern for the reference of future planning, that is, at least 30% water use efficiency improvement and 20% pollutant emission reduction are necessary to achieve the goal of the water quality target, and moderate socio-economic development is conducive to regional sustainability.

To further discuss the annual change of water quality under the H-M scenario, Figure 8 shows the spatial distribution of COD and ammonia nitrogen concentration on the 30th, 120th, 210th and 300th days of the simulation period.

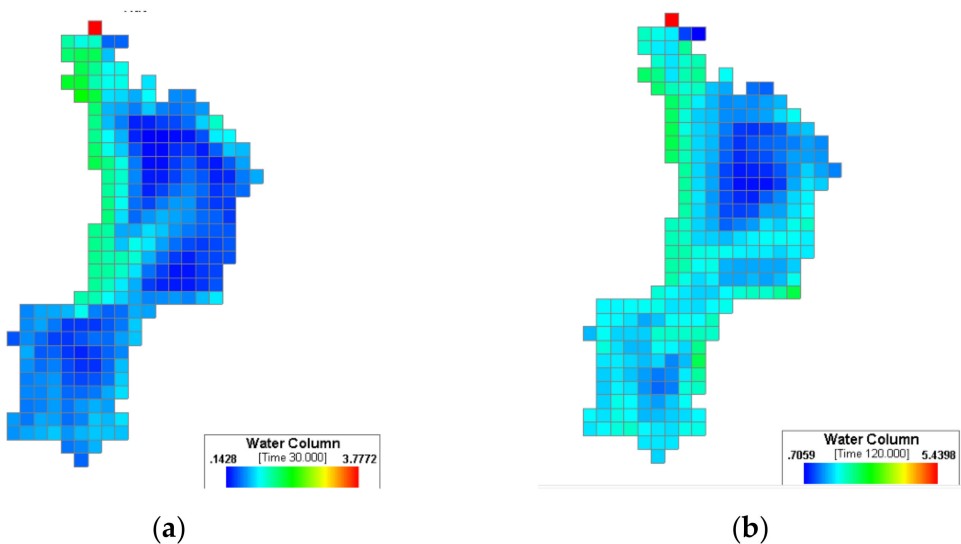

(**a**)                       (**b**)

**Figure 8.** *Cont.*

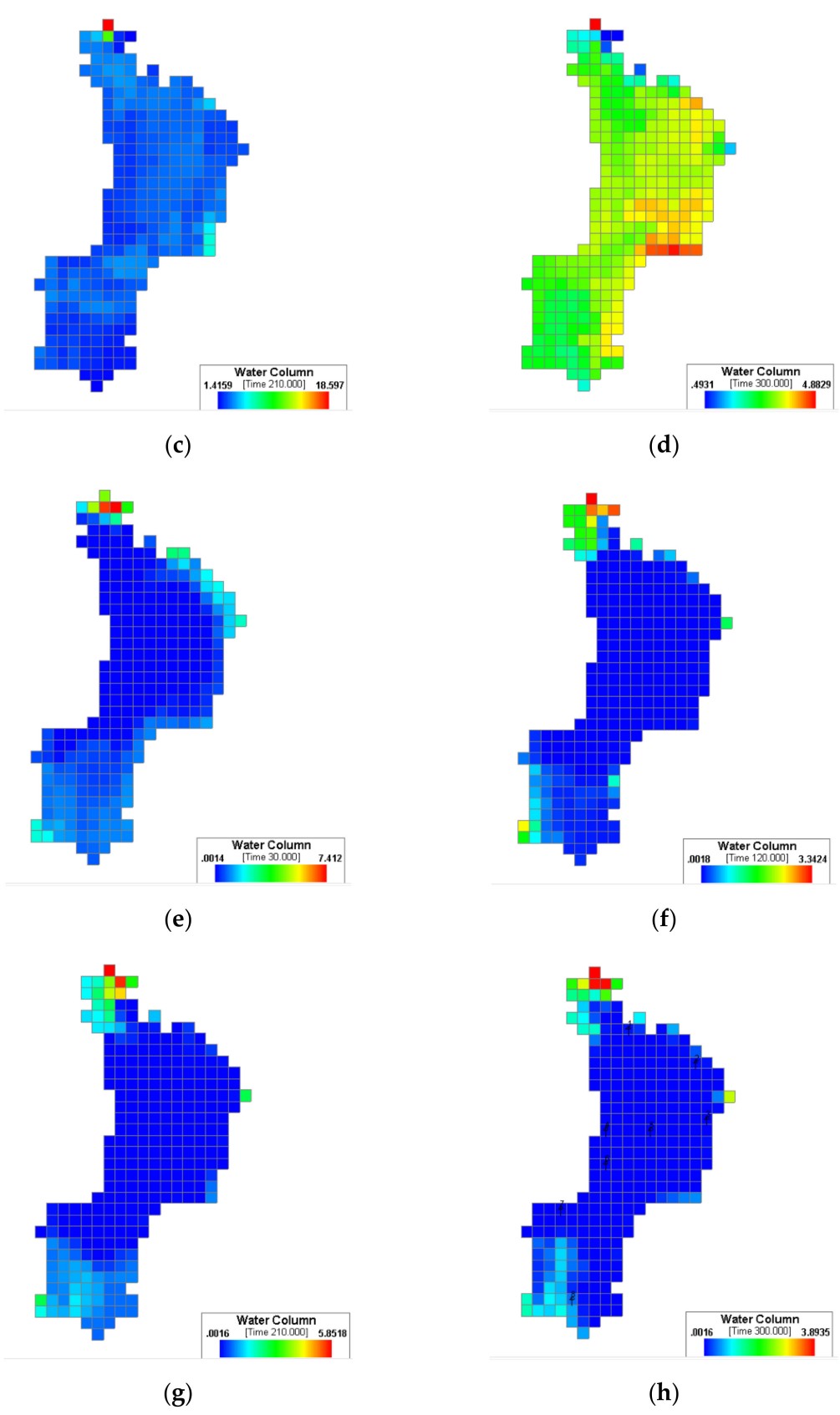

**Figure 8.** COD concentration distribution on the 30th (**a**), 120th (**b**), 210th (**c**) and 300th (**d**) days of the simulation period and ammonia nitrogen concentration distribution on the 30th (**e**), 120th (**f**), 210th (**g**) and 300th (**h**) days of the simulation period under the H-M scenario.

It can be seen that the change trend of the COD concentration basically starts from the northwest of the lake, that is, the part near the main urban area of Kunming City. From January to April, due to the increase of rainfall intensity, the increasing COD emission entering the lake diffuses from all estuaries to the whole lake, especially the west and north sides, showing a phenomenon of high concentration in the estuary area and low concentration in the central area. In July, COD concentration in most areas of the lake is stable at the highest value. By October, due to the reduction of COD emission into the lake, the COD concentration in the central area will be higher than that in the estuary area, especially in the northwest area. As for the ammonia nitrogen, it can be seen that in all four seasons of the year, the spatial distribution of ammonia nitrogen shows a phenomenon of high concentration in the estuary area and low concentration in the central area. As the transformation rate of ammonia nitrogen in the lake is much faster than that of COD, the concentration of ammonia nitrogen will decrease rapidly with the increase of diffusion distance, which make it lower in most areas of the lake. The northwest area of the lake is close to the city with a high pollution load, so the concentration in this area is the highest. Overall, the year-round COD and ammonia nitrogen concentrations are satisfactory, which further confirms the feasibility of the H-M scenario.

## 4. Conclusions

In this study, an integrated approach for evaluation of the coupling coordination degree (CCD) between socio-economic development and environmental protection in the Dianchi Lake basin was established. This model can not only propose a development pattern as a reference for future planning, which would be helpful for continuously improving the regional sustainability, but also be applied in other watersheds, helping to calculate their coordination degree.

After research, it can be found that the High protection–Medium development (H-M) scenario performed best in coordination, and the EFDC analysis further confirmed that the water quality under this scenario had the most significant improvement. Therefore, the H-M scenario is the most suitable pattern as a reference for future planning, that is, at least 30% water use efficiency improvement and 20% pollutant emission reduction are necessary to achieve the water quality target, and moderate socio-economic development is conducive to regional sustainability. Policy makers can achieve the key indicators set by this scenario through the following methods.

In terms of macro-economy, the expected development speed should be reduced to 9.5% and the population growth rate should be controlled at approximately 1.9%, which can be achieved by industry structure regulation, construction of satellite town and population redistributing in the main urban area. The growth speed of industrial development should be maintained at approximately 10.7%, by eliminating backward technology enterprises, accelerating technological upgrading and establishing industry admittance. Since most of the pollution sources in the agriculture and service sectors are not equipped with collection and treatment facilities, their scale most be strictly controlled, by adjusting the layout of agricultural production or reducing the planting area of crops and flowers.

As for pollutant reduction, the amount of COD emission should be reduced by 18% and that of ammonia nitrogen emission should be decreased by 22%. The specific implementation methods include: reducing the rural non-point source pollution by using environmental protection fertilizers, continuous planting and other emission reduction measures; reducing urban sewage discharge by improving facilities and eliminating high water consumption enterprises; improving the sewage treatment standard; and increasing the sewage treatment rate to approximately 98% to reduce the output of pollution load, etc.

In addition, further improvement of system coordination can be achieved according to the weight of the indicators. On the one hand, it is possible to reduce the development level of the socio-economic subsystem by reducing the economic growth rate in the long run; on the other hand, a more effective measure to do this is to increase the development level

of the water environment subsystem by improving water supply pipelines and increasing wastewater reuse rates to reduce the total amount of urban sewage.

There are still some deficiencies in this article, such as a short time range. Further research in the future can start from two aspects: expanding the time scale and increasing the number of indicators, such as other forms of water pollutants.

**Author Contributions:** Conceptualization, S.Z., H.K. and X.D.; methodology, H.K.; validation, H.K.; formal analysis, H.K.; investigation, H.K. and Y.L.; resources, H.K., Y.L. and S.Z.; writing—original draft preparation, H.K.; writing—review and editing, H.K.; visualization, H.K.; supervision, S.Z.; project administration, S.Z.; funding acquisition, S.Z. All authors have read and agreed to the published version of the manuscript.

**Funding:** This research was funded by National Natural Science Foundation of China, grant number 51978374.

**Acknowledgments:** Thanks to Kunming Institute of ecological and Environmental Sciences for helping to collect data of this research.

**Conflicts of Interest:** The authors declare no conflict of interest.

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
