# Peer review of "Quantification of the Coordination Degree between Dianchi Lake Protection and Watershed Social-Economic Development: A Scenario-Based Analysis"

_sustainability, doi:10.3390/su13010116_

Round 1
Reviewer 1 Report
The concept of the paper is interesting and shows also good resultats.
But generally and also in this work for building a good and suitable scenarios were better to use phosphorus like indicator.
Phosphorus it's a very important eutrophication factor and a conservative pollutant.
In Water quality simulation under typical scenarios you wrote:
„As the transformation rate of ammonia nitrogen in the lake is much faster than that of COD, the concentration of ammonia nitrogen will decrease rapidly with the increase of diffusion distance, which lead to a low value of it in most areas of the lake.“
But is not clear whats happend with ammonia,
Perhaps were better to use total nitrogen for calculation?
However in this study ist not clear what are the suorce and whats happens with COD and ammonia in the Dianchi Lake.
The Conclusion is superficial and insufficient.
You wrote:
„The objective of this study is to find the most suitable development scenarios and take them as a reference for the preparation of future development plans of Dianchi Lake basin. „
But for better understanding were necessary to give more information and to explain how can planners use the results.
Figure 8: chart its bed and text ist not redable.
Reviewer 2 Report
Article
Quantification of Coordination Degree between Dianchi Lake Protection and Watershed Social Economic Development: A Scenario-Based Analysis
by Hansheng Kong , Yilei Lu, Xin Dong, and Siyu Zeng
Title, Abstract and key words
- Postal address missing
- Choose key words that differ from words used in the title
Introduction
- In text citations need adjustment to the Sustainability style
- Ideas are revealed in a discontinuous manner
- Please define ‘coordination’ as you appreciate and use the word in the manuscript
- lines 40, 43, 45: reference required
Study area
- provide characteristics of the lake under study: area, depth, hydrology, meteorology, hydrochemistry
Coordination model construction
- provide basics description of the CCD model
- Table 1 : explain meaning of ‘indicators’ and ‘weight’ of indicators
- GDP: explain
- line 128 reference required
- explain COD
Scenarios definitions
- Table 2 : explain how the indicators were selected
EFDC model
- more references required
- in text citations require correction
Results and discussion
- explain meaning and values of the ‘e’ and ‘s’ indexes
- Fig.5, 6, 7 : explain x-axis: COD values are way too high, Explain Huiwan Middle, ….; explain why concentrations in Janury and December are different by an order of magnitude or more- after all January of 2026 follows December of 2015
References
- word-wide historical and up to date references are missing
Reviewer 3 Report
Observations
Lines 60-62: Announce Figure 1 before the figure.
Line 91: After “equation 1” replace “.” with “:”.
Line 92: Not indented and with small letter: “where”.
Same observations as for lines 91 and 92 are applicable whatever you introduce equations and explain variables. E.g. lines 103 and 104, etc.
Line 113 and further: all variables must be written italic in equations and in text.
Figure 2: Use the same scale for the vertical axis, it means from 0.30 to 1.10, for all graphical representations.
Line 324: “Tto” is “To”.
Lines 358-362: Move this paragraph at the beginning of section 4.
Slightly develop section Conclusions.
! Good paper! Congratulations!
Round 2
Reviewer 2 Report
Dear Authors,
I found the revised manuscript improved.
My disposition is 'accept'.
Author Response
Thank you again for taking your time to review this manuscript!We really appreciate it.